# Analysis of Genetic Diversity and Population Structure of Cowpea (*Vigna unguiculata* (L.) Walp) Genotypes Using Single Nucleotide Polymorphism Markers

**DOI:** 10.3390/plants11243480

**Published:** 2022-12-12

**Authors:** Mbali Thembi Gumede, Abe Shegro Gerrano, Assefa Beyene Amelework, Albert Thembinkosi Modi

**Affiliations:** 1Centre for Transformative Agricultural and Food Systems, School of Agricultural, Earth and Environmental Sciences, College of Agriculture, Engineering and Sciences, University of KwaZulu-Natal, Private Bag X01, Scottsville, Pietermaritzburg 3209, South Africa; 2Agricultural Research Council—Vegetables, Industrial and Medicinal Plant Institute, Private Bag X293, Pretoria 0001, South Africa; 3Department of Plant Sciences and Plant Pathology, Montana State University, Bozeman, MT 59717-3150, USA

**Keywords:** cowpea, genetic diversity, population structure, single nucleotide polymorphism

## Abstract

Cowpea (*Vigna unguiculata* (L.) Walp) is an important legume crop with immense potential for nutritional and food security, income generation, and livestock feed in Sub-Saharan Africa. The crop is highly tolerant to heat and drought stresses which makes it an extremely important crop for improving resilience in crop production in the face of climate change. This study was carried out to assess the genetic diversity and population structure of 90 cowpea accessions using single nucleotide polymorphism (SNP) markers. Out of 11,940 SNPs used, 5864 SNPs were polymorphic and maintained for genome diversity analysis. Polymorphic information content (PIC) values ranged from 0.22 to 0.32 with a mean value of 0.27. The model-based Bayesian STRUCTURE analysis classified 90 cowpea accessions into four subpopulations at K = 4, while the distance-based cluster analysis grouped the accessions into three distinct clusters. The analysis of molecular variance (AMOVA) revealed that 59% and 69% of the total molecular variation was attributed to among individual variation for model-based and distance-based populations, respectively, and 18% was attributed to within individual variations. Furthermore, the low heterozygosity among cowpea accessions and the high inbreeding coefficient observed in this study suggests that the accessions reached an acceptable level of homozygosity. This study would serve as a reference for future selection and breeding programs of cowpea with desirable traits and systematic conservation of these plant genetic resources.

## 1. Introduction

Cowpea (*Vigna unguiculata* (L.) Walp, 2n = 2x = 22) is an important legume crop that belongs to the genus Vigna, family Fabaceae, and order Fabales, with a genome size of 620 million base pairs [1]. Cowpea is a herbaceous annual plant widely grown in tropical and subtropical regions of Sub-Saharan Africa, East Asia, and other developing countries [2,3]. The crop plays a major role in both human and animal nutrition and food as well as food security and income generation for farmers and agro-traders [3]. In addition to the crop’s importance in sustaining food security, cowpea possesses good resilience to extreme heat and drought conditions which is extremely important in improving the resilience of the crop to the current climate change [4,5].

Moreover, cowpea has significant importance to cropping systems on account of its ability to grow in low fertility soils, as a complementary crop in rotation with cereals to break the life cycle of pathogens of cereals infested in the soil and consequently improve the fertility of the soil [6]. Globally, the cowpea average yield ranges from 0.1 t ha^−1^ to 0.59 t ha^−1^ which is lower than its expected potential yield of 1.5 t ha^−1^ to 3 t ha^−1^ under suitable environmental conditions [7]. This is caused by the narrow genetic base of improved varieties and their susceptibility to abiotic and biotic stresses. It is of high importance to develop improved varieties of cowpea to increase productivity in order to help alleviate poverty in Sub-Saharan Africa and also to meet the market demand. The production of improved cowpea varieties for traits such as high yield and nutritional status will largely benefit both subsistence and commercial farmers. Therefore, characterizing genetic diversity in any crop species is important for optimal germplasm utilization, conservation, and crop improvement programs. In support, Kondwakwenda [8] also stated that the availability of appropriate genetic diversity is imperative for the sustenance and success of any crop breeding program.

The assessment of the genetic diversity of a particular crop is achieved using morphological, biochemical, and molecular markers [3,9]. Although biochemical markers are more reliable than morphological markers, they are both reported to be influenced by environmental factors. These markers provide genetic diversity information on the basis of genotype performances using agronomic traits and may differ at different stages of growth and development as well as in the growing environment. This may hinder the real genetic variation among genotypes [10,11] and reduce the accuracy of the results obtained. Hence, there was an evolution of the development of DNA molecular markers [12,13,14,15]. Molecular markers are neutral to environmental effects and the genetic diversity is reviewed at the genomic DNA level; therefore, it is helpful to envision the precise genetic diversity among genotypes [3,16]. However, they may not be associated with any agronomic traits and needs to be supplemented with morphological markers in order to infer a meaningful conclusion.

There are several DNA markers that have been developed to determine the genetic diversity of cowpea. These include restriction fragment length polymorphism (RFLP) [16,17], simple sequence repeats (SSR) [18], single nucleotide polymorphism (SNP) [19], amplified fragment length polymorphic (AFLP) [20], and random amplified polymorphic DNA (RAPD) [21]. In recent advances in molecular genetics and molecular biology, the use of SNP markers has emerged to be the most preferred molecular marker because of its high genomic abundance, cost-effectiveness, reliability, and ease of application in comparison to other polymerase chain reaction (PCR)-based molecular markers [22,23]. Hence, the SNP markers were used in the current study. The study by Desalegne [24] compared the efficiency of SNP and SSR marker-based analysis of genetic diversity in 95 cowpea accessions collected from East Africa and the International Institute of Tropical Agriculture (IITA) inbred lines. Their study revealed that SNP markers were found to be more effective than SSR markers to determine the association between cowpea varieties; hence, the study suggested the utilization of SNP markers in the future analysis of genetic diversity and population structure in cowpea. Similarly, the recent study by Nkhoma [19] evaluated the genetic diversity in 90 cowpea genotypes using SNP markers and phenotypic traits and the study showed that SNP markers were more efficient in differentiating the diversity among and within the cowpea genotypes evaluated.

There are sequencing technology-based tools, which is next-generation sequencing (NGS), that have emerged to discover SNP markers, using genotyping-by-sequencing (GBS), which has been reported to be efficient, inexpensive, and fast developing in sequencing plant genomes [25,26]. In addition, a new type of marker known as diversity arrays technology (DArT) has been recently established for genotyping and genome sequencing needlessly of sequence information [3]. In cowpea, the DArT marker has been recently used by Gbedevi [3] to study the genetic diversity and population structure of 498 cowpea accessions collected from the Republic of Togo and their study revealed the presence of four major clusters among the accessions studied and the accessions were not clustered according to the regions where they were collected suggesting that the clustering did not closely resemble the geographical areas of the collections. Classic DArT markers have been substituted by DArTseq markers based on GBS. DArTseq and SNP markers based on GBS technology have been successfully applied in different crops including legumes such as cowpea [3,27], chickpeas [28], common beans [29], and Bambara groundnut [30]. Hence, the objective of this study was to assess the magnitude of the genetic diversity and population structure among cowpea genotypes using single nucleotide polymorphisms (SNP) markers.

## 2. Results

### 2.1. Allele Polymorphism

SNP distribution per chromosome and the gene diversity parameters measured from 90 cowpea accessions are presented in Table 1. The genetic diversity parameters analysis was conducted using 5864 (49%) SNPs that remained after filtering out monomorphic and minor allele frequencies of less than 2%. The number of polymorphic SNPs per chromosome ranged from 345 on chromosome 1 to 668 on chromosome 3 with an overall mean of 488 per chromosome. The proportion of polymorphic SNPs per chromosome varied from 31.82% on SNPs of unknown chromosome origin to 57.35% SNPs on chromosome 9, with an overall mean of value of 49.11% per chromosome. The mean number of effective allele (Ne) per chromosome was the highest on chromosome 7 (1.48 ± 0.013), followed by chromosome 10 (1.47 ± 0.013) and chromosome 4 (1.47 ± 0.013) whilst the lowest values were observed on chromosome 9 (1.34 ± 0.014) and chromosome 8 (1.37 ± 0.013). The observed heterozygosity ranged from 7.6% to 9.6% with a mean value of 8.4%. The unbiased gene diversity (uHe) values ranged from 0.221 to 0.291 per chromosome with an overall mean of 0.267. The fixation index (FIS) values ranged from 64% on chromosome 11 to 72% on chromosome 7 with a mean value of 67%. The mean polymorphic information content (PIC) value was 0.27, in which the PIC values per chromosome ranged from 0.22 to 0.32.

### 2.2. Population Structure and Clustering

The population structure of the 90 accessions was examined using model- and distance-based structure analyses. At K = 2, the first cluster consisted of 82 genotypes, of which 29% were admixtures while the second cluster contained 8 genotypes in which majority of them (62%) were admixture. At K = 3, the first cluster contained 62% of the cowpea genotypes and 68% were admixtures. The second and the third clusters consisted of 6 and 28 genotypes with 67% admixtures each, respectively (Figure 1). However, the STRUCTURE analysis estimated that the most suitable number of subpopulations was at K  =  4 (Appendix A), indicating that the 90 accessions could be grouped into four subpopulations (SP1–SP4) based on differences in their genetic makeup (Figure 1). SP4 (red) contained only five accessions with admixtures from SP2 and SP3. SP2 (yellow) contained 40 accessions that share more admixture membership with the SP1, SP3 and SP4. P3 (green) contained 27 accessions, which share admixture membership with the other three subpopulations. SP4 (blue) contained 18 accessions with admixture from the other subpopulations. The admixture level in the four subpopulations ranged from 70% to 80%, which indicated that these subpopulations shared more admixture memberships. Individuals with a probability score of above 90% for a given cluster were considered as ‘pure’, whereas those with less than 90% were labeled as ‘admixture’.

However, the distance-based cluster analysis generated using Nei’s genetic distance using a neighbor-joining algorithm revealed the presence of three distinct clusters in the population represented by the 90 accessions (Figure 2). The clustering patterns of the two approaches were similar and constituted similar sets of accessions. For example, C1 (red) contained 41 accessions, 5 accessions from SP1, 27 from SP4, and 9 from SP3 of the STRUCTURE-generated clusters. C2 (black) consisted of 24 accessions, of which 18 accessions from SP2 and 5 were from SP3. C3 (blue) contained 26 accessions all from SP3. The discrepancy between the two clustering approaches could result from admixtures since only 43% of the tested genotypes were considered pure. The clustering patters did not match the geographic origins of the accessions. C1 consisted of 95% South African, 2% Nigerian, and 2% Tanzanian accessions. The majority of the accession (92%)) clustered in C3 were collected from South Africa and 4% from Nigeria and Tanzania each. However, C2 was dominated with accessions from South Africa (52%), Nigeria (39%), and Kenya (4%).

### 2.3. Genetic Diversity among Subpopulations

The population genetic diversity estimates on 90 accessions were analyzed based on the four subpopulations generated by STRUCTURE and three populations generated by DARwin (Table 2). SP3 revealed the highest values for most of the genetic parameters and displayed the highest level of genetic diversity (He = 0.247 and I = 0.381). SP1 displayed the lowest level of genetic diversity (He = 0.162 and I = 0.189) but 68.4% of the loci were fixed (Table 2). SP3 had the highest number of private alleles (508) and the highest percentage of polymorphic loci (91%). Based on the three subpopulations generated by cluster analysis, C2 revealed the highest genetic diversity for most of the genetic parameters except for the fixation index. C1, on the other hand, revealed the lowest genetic diversity for all the studied genetic parameters. In C1, 68% of the alleles were fixed and 87% of loci were polymorphic.

### 2.4. Analysis of Molecular Variance (AMOVA)

AMOVA was performed among subpopulations estimated from both model-based and distance-based populations (Table 3). The results of AMOVA in model-based populations indicated that the majority of the variance occurred among individuals within populations and accounted for 59.4% of the total variation. However, 18.4% and 22.1% of the total variation was attributed to differences within individuals and among populations, respectively. Similarly, in the distance-based population, the majority of the variance was observed among individuals within the population and accounted for 68.9% of the total genetic variance. The mean fixation index within individuals was significantly high and positive in all classes of populations suggesting that outcrossing among the tested cowpea populations was low. The variation existed among populations was positive and significant suggesting that these populations were highly differentiated. Similarly, the relatively low level of variation observed within individuals was attributed to the high fixation index value.

Genetic differentiation (F_ST_) estimates among the subpopulations ranged from 0.103 between SP3 and SP4 to 0.239 between SP1 and SP2. The gene flow ranged from 0.8 (between SP1 and SP2) to 2.2 (between SP3 and SP4). The genetic distance among populations ranged from 0.089 between SP3 and SP4 to 0.214 between SP1 and SP2. The genetic identity (GI) ranged from 0.81 between SP1 and SP2 to 0.92 between SP3 and SP4 (Table 4). According to Wright [31] standard guidelines for the interpretation of genetic differentiation, all pairs of subpopulations showed a moderate level of population differentiation. However, SP1 showed a relatively higher degree of differentiation (0.239) from the rest of the subpopulations. Gene flow among the subpopulation was relatively high between SP2, SP3, and SP4 (Table 4). The observed high genetic differentiation among the subpopulations could be explained by the low gene flow among subpopulations.

## 3. Discussion

The analysis of genetic diversity in crops is a prerequisite for the success of any plant breeding program [32]. Therefore, assessing the population structure and genetic diversity of crops is fundamental to implementing efficient genetic resource management and conservation strategies. The application of high-throughput molecular markers provides a better understanding of genomic diversity and the population structure of germplasm and can speed up the identification of superior groups for further hybrid development [33]. The current study used 5864 SNP markers to assess the pattern and level of genetic variation and genetic structure among 90 cowpea accessions collected from four geographic origins. 

The quality and the discriminatory power of a given marker system are assessed by its PIC values [34]. It is important to note that SNP markers are bi-allelic in nature, hence their PIC values are restricted to 0.5, which is considered to be low or moderately informative as compared to SSR markers [35]. The mean PIC value of 0.27 reported in this study agreed with Gbedevi [3] who reported a PIC value of 0.25 but relatively higher than the one reported by Sodedji [36] (PIC = 0.22). The results suggest that the SNP markers that were used in this study showed a moderate level of polymorphism and revealed the existence of genetic diversity among the tested genotypes. The number of SNP markers used and the number of accessions studied might explain the observed differences among the reported PIC values and allelic polymorphism in this study and other previously reported studies. Kondwakwenda [8] also indicated that the observed variation in the quality and performance of SNP markers in different studies depends on the number of accessions studied, the type of markers used and the type of germplasm studied. Nonetheless, the SNP markers used in this study were relatively informative and reliable in assessing the diversity of the cowpea accessions studied.

The mean number of effective alleles per locus reported in this study was 1.43, which was comparable to the 1.41 reported by Fatokun [37]. The gene diversity was further expressed using the probability of observed (Ho) and expected (He) heterozygosity, which was the true indicator for the degree of genetic variation within and among the populations assessed. The average Ho and He in the present study were 0.075 and 0.202, respectively, for the 90 accessions which were comparable to the Ho of 0.05 and He of 0.31 reported by Gbedevi [3] for 70 cowpea accessions. Xiong [38], on the other hand, assessed 768 worldwide cowpea germplasm collections maintained at USDA GRIN and reported a Ho value of 0.06 and a He value of 0.35. Nonetheless, the He was moderately low in this study but generally higher than the Ho for all subpopulations. Fatokun [37] also reported a similar trend in cowpea. Govindaraj [39] alluded that the low observed heterozygosity suggests a high level of inbreeding within the subpopulations. 

A moderate fixation Index (F_IS_ = 67%) was observed in this study indicating 67% of the SNP loci used were fixed in the studied accessions. However, a relatively high (F_IS_ = 0.83) was reported by Gbedevi [3]. The low observed heterozygosity and the relatively high rate of fixation index exhibited by the populations were explained by the fact that cowpea is a self-pollinated crop possessing a low out-crossing rate and low within-accession variability. Although the outcrossing rate in cowpea is low and ranges from less than 0.15 up to 1.58% depending on the genotypes involved and the environment, where it is grown [40], further purification (self-pollination) of the accession is needed. The self-pollination nature and low outcrossing rate of cowpea have been reported to be the major contributor to the observed low genetic variation among cowpea accessions [6,41].

Population structure analysis is the key to assessing the genetic structure of a given population and the basis for complex marker-trait association analysis [42]. The model-based population structure analysis grouped the 90 accessions into four subpopulations based on the peak of delta K (∆K) at K = 4. The admixture level ranges from 50% in SP4 to 60% in SP1. The high proportion of admixture detected indicates either these subpopulations share the same ancestral progenitor or there was gene flow between the subpopulations. A similar trend has been noted on the clustering based on the geographic origins of the accessions whereby accessions were not clustered together as per the origin. The majority of the accessions in C1 and C3 were from South Africa then Nigeria and Tanzania while C2 was dominated with accessions from South Africa, Nigeria, and Kenya. This could be due to formal or informal seed exchange from among farmer and traders. The distance-based cluster analysis using the neighbor-joining method showed the presence of three distinct clusters, which was not consistent with the results of the structure analysis. However, the pattern and the number of accessions maintained in each clustering approach were similar. The differences observed between the model based on STRUCTURE analysis and distance-based cluster analysis in the size and number of subgroups can be explained by the presence of admixtures within the subpopulations. In both model-based and distance-based clustering approaches, the grouping patterns were inconsistent with the growth habit and geographic origins of the studied accessions. Similar phenomena whereby subpopulations were grouped irrespective of the grouping criteria have also been reported by other researchers studying the genetic diversity in cowpea accessions [3,36,43]. In contrast, Ravelombola [44] reported clustering of genotypes based on growth habit resulted in two highly differentiated subpopulations. 

The seed size and seed coat color preference can highly influence the genetic diversity in cowpeas [5]. Classification based on growth habits and other agronomic traits such as seed shape and seed coat color is significantly influenced by the breeding programs because these traits have been used to classify genotypes. Qualitative traits such as growth habits, seed size, seed shape, and seed coat color are also important traits for farmers and consumers preferences [45]. Therefore, it is important to incorporate farmers’ and consumers’ preferred traits in future selections to enhance varietal adoption among farmers. In the UPGM clustering, C1 (red) was the highest group comprising of 41 (46%) genotypes with brown (22%), red (12%) and cream (24%) seed coat color. Similarly, 23 accessions were grouped in C2 (black) comprises of 35% of brown seed coat color and 26 accessions grouped in C3 (blue) comprised of 35% black seed coat and 19% brown seed coat. In addition, the majority of accessions clustered in C1 and C2 were kidney-shaped while C3 was dominated with rhomboid-shaped accessions. In terms of phenotypical variation, the erect and prostrate type were not different with respect to seed shape. The study by Hellens [46] in peas reported the lighter seed coat color as a human preference during domestication. 

Similarly, C1 was dominated by erect (32%), while C2 was dominated by accessions with unknown growth habits (44%). C3 was mainly dominated by the prostate (54%) growth habit type. Regarding the wide distribution of cowpea, accession was studied based on growth habit, the prostrate and erect types were more dominant than the semi-erect and climbing types. Growth habit is a morphologically important qualitative trait in cowpea production that highly affects crop yield and tillage method and further defines the shape of the plants and dictates how the plant should be harvested [47]. Plant growth habit has been a major breeding target for crop improvement. Therefore, determining the genetic mechanisms that control the plant type will assist in cowpea growth development improvements. However, the results revealed that growth habits could not be used as an index for evaluating genetic diversity and for genotype classifications. The study by Khan [48] on Bambara groundnut has also reported the same findings. 

The AMOVA results revealed that the majority of the total molecular variation (59% and 69%) was due to differences among individuals within a population, 22% and 12% of the variation were attributed to the difference among the population, and 18% and 19% was due to variation within individuals. In the model-based approach, the among-population variation was higher than the within individuals variation. The magnitude of variations among and within populations was further quantified by genetic differentiation observed among the populations (F_ST_ = 0.221). The results indicate high genetic differentiation between four subpopulations based on the standard guideline of Wright [31]. The studies by Gbedevi [3] also reported a high genetic differentiation (F_ST_) value of 0.423 between two major reported subpopulations of 498 cowpea accessions. However, Sarr [43] also reported a low genetic differentiation ranging from 0.018 to 0.100. The differences reported in genetic differentiations could be attributed to the diversity and number of accessions used and the number of markers involved to assess the genetic diversity.

The reported level of genetic differentiation reported in this study could be explained by the gene flow among subpopulations [49]. The gene flow among the studied populations ranged from 0.796 to 2.172 and according to Wright [50], where gene flow < 1 indicates limited gene exchange among population. This result suggested that a moderate gene flow occurred in this study and led to high genetic differentiation between the populations. Furthermore, genetic distance was used to measure the relatedness between individuals in a population. In this study, the low pairwise genetic distance was observed ranging from 0.089 to 0.214 revealing wide genetic variations among the tested cowpea accessions. This result suggests that the accessions studied are unique and have greater potential to contribute to new varieties for breeding programs in South Africa. The understanding of genetic diversity among cowpea populations studied in this study will improve the subsequent planning in future cowpea breeding and contribute useful information in conservation and managing genetic diversity required for the vigorous and successful breeding program.

## 4. Materials and Methods

### 4.1. Plant Materials

The study evaluated the genetic diversity of 90 cowpea accessions sourced from the Agricultural Research Council—Vegetables, Industrial and Medicinal Plants (ARC-VIMP) gene bank, Pretoria, South Africa. These accessions were collected from different parts of Africa including South Africa, Tanzania, Kenya, and Nigeria. The geographic origin and growth habits of the accessions are presented in Appendix A.

### 4.2. DNA Extraction and Sequencing

The cowpea genotypes were grown in a seed germination chamber at the Biosciences eastern and central Africa International Livestock Research Institute (BecA-ILRI) hub in Nairobi, Kenya for genotyping. Ten-day-old leaf materials were sampled from the seedlings and the leaf samples were frozen in liquid nitrogen and stored at −80 °C for genotyping. DNA extraction was done using a NucleoMag Plant DNA extraction kit from Takara Bio USA, Inc. The genomic DNA extracted was in the range of 50–100 ng/μL. DNA quality and quantity were checked on 0.8% agarose gel. 

Libraries were constructed according to Killian [51] using the DArTSeq complexity reduction method through digestion of genomic DNA using a combination of PstI and MseI enzymes and ligation of barcoded adapters and common adapters followed by PCR amplification of adapter-ligated fragments at the Biosciences Eastern and Central Africa hub of the International Livestock Research Institute (BecA-ILRI) in Nairobi. Libraries were sequenced using Single Read sequencing runs for 77 bases. Next-generation sequencing was carried out using Hiseq2500. DArTseq markers scoring was achieved using DArTsoft14, which is an in-house marker scoring pipeline based on algorithms. Two types of DArTseq markers were scored, SilicoDArT markers and SNP markers, which were both scored as 1 for presence, 0 for absence, and 2 for heterozygotes of the restriction fragment with the marker sequence in genomic representation of the sample. The DArTseq markers were scored using 11,940 SNP markers, which were set to 11 chromosomes of cowpea. The SNP markers were aligned to the cowpea reference genome, Vunguiculata_469_v1.0 to identify chromosome positions.

### 4.3. Data Analysis

A total of 90 cowpea accessions were genotyped with 11,940 SNP markers. Monomorphic and SNPs with a minor allele frequency of less than 2% were filtered out and 5864 (49%) SNPs were retained for further analysis. Genotypic data were subjected to analyses of molecular variance (AMOVA) and various measures of genetic diversity within and among inferred subpopulations using GenAlex software version 6.5 [52]. Genetic diversity parameters such as the number of effective alleles per locus (Ne), Shannon’s Information Index (I), gene diversity (He), and the polymorphic information content (PIC) were determined using the protocol of Nei and Li [53] using GenAlex software version 6.5. The genotypic data were used to obtain a dissimilarity matrix using the Jaccard index as described by Debener [54]. The matrix was then used to run a cluster analysis based on a neighbor-joining algorithm using the un-weighted pair group method with arithmetic average (UPGMA) in DARwin 6.0 software [55]. Bootstrap analysis was performed for node construction using 10,000 bootstrap values.

The Bayesian genotypic clustering approach of STRUCTURE 2.3.4 [56] was used to determine the population structure. An admixture model with independent allele frequencies, without prior population information, was used to simulate the population. The STRUCTURE program was set as follows: a burn-in period length of 100,000, and after burn-in, 100,000 Markov Chain Monte Carlo (MCMC) were used. This model assumes that the genome of each individual is a mixture of genes originating from K unknown ancestral populations. For joint inference of the population substructure, K ranging from 2 to 7 was set up, with ten independent runs for each K. The most probable value of K for each test was detected by ΔK [57] using the STRUCTURE HARVESTER [58]. Each individual genotype was grouped into a given cluster using the ‘membership coefficient’ for each cluster interpreted as a probability of membership. The genotype membership was determined by the computer program CLUMPP [59].

## 5. Conclusions

The current study revealed the existence of genetic diversity among and within the cowpea accessions analyzed and showed the effectiveness and reliability of SNP markers. The study revealed that 49% of the selected SNP markers were highly polymorphic and efficiently discriminate the tested cowpea accessions. The low heterozygosity and the high inbreeding coefficients observed among cowpea varieties indicate that the accessions reached an acceptable level of homozygosity. The model-based (structure analysis) and distance-based (UPGM) clustering approaches were used in this study. The model-based analysis revealed the presence of four subpopulations at K = 4 whereas the distance-based cluster analysis classified the cowpea accessions into three distinct clusters. The subpopulations identified exhibited the high level of genetic diversity and were moderately differentiated. These subpopulations could serve as heterotic groups and a relevant source of genes for future breeding and selection of diverse cultivars with different traits. Therefore, the results obtained from this study will be highly valuable for the development of new varieties adapted to diverse environments. Consequently, this study will substantially contribute to the utilization, conservation, and broadening of the use of genetic resources of cowpeas for future improvement. 

## Figures and Tables

**Figure 1 plants-11-03480-f001:**
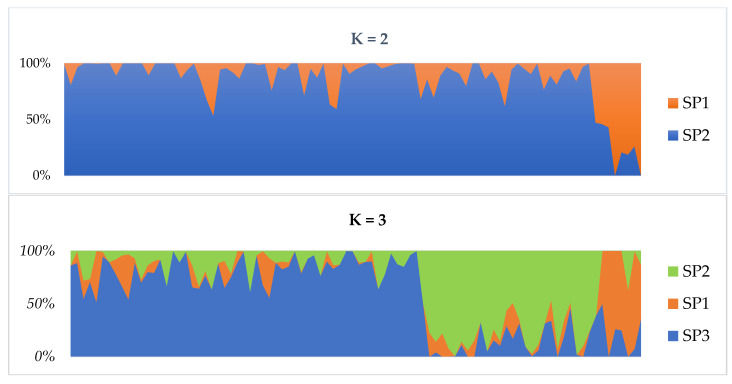
Population structure analysis using a Bayesian-based approach. Population structure analysis of 90 cowpea accession from K = 2 to K = 4 based on inferred ancestry (Q matrix).

**Figure 2 plants-11-03480-f002:**
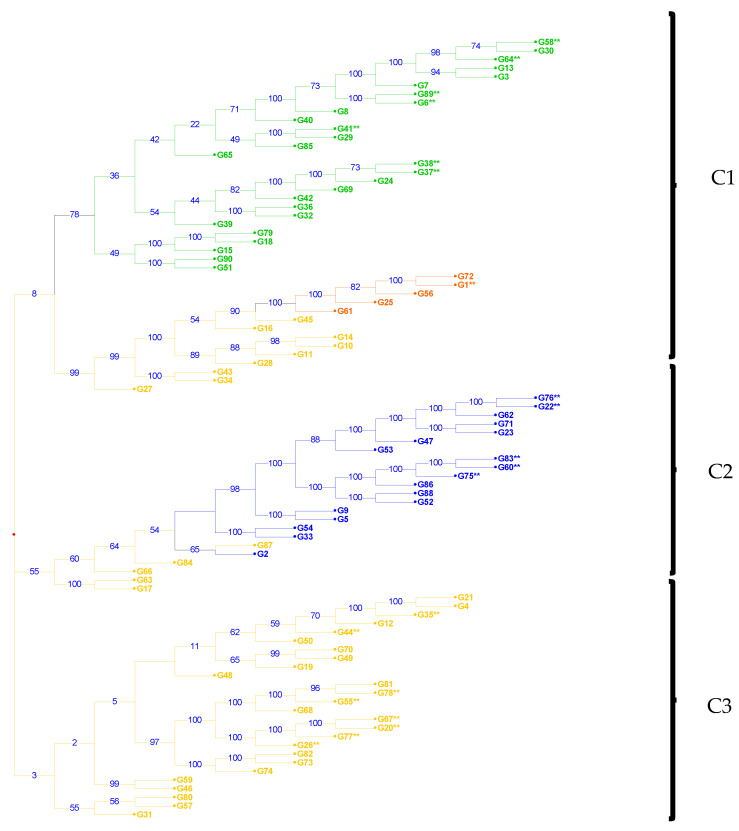
Neighbor joining analysis showing the genetic relationships among 90 cowpea accession tested using 5864 SNP markers. The different colors indicate the clustering generated by STRUCTURE analysis: SP1 = red, SP2 = orange, SP3 = green, and SP4 = blue. ** represents the pure varieties identified in STRUCTUR analysis.

**Table 1 plants-11-03480-t001:** Genetic diversity within and among 90 cowpea accessions genotypes based on 5864 SNPs markers.

Chromosome	NSU	NPS	%P	N_e_	H_o_	uH_e_	F_IS_	PIC
1	670	345	51.49	1.43 (0.017)	0.088 (0.004)	0.269 (0.008)	0.66 (0.012)	0.27 (0.008)
2	725	377	52.00	1.41 (0.016)	0.077 (0.003)	0.258 (0.008)	0.68 (0.011)	0.26 (0.008)
3	1222	668	54.66	1.46 (0.012)	0.082 (0.003)	0.288 (0.006)	0.70 (0.008)	0.29 (0.005)
4	1123	533	47.46	1.47 (0.013)	0.094 (0.004)	0.290 (0.006)	0.67 (0.010)	0.29 (0.006)
5	965	453	46.94	1.42 (0.015)	0.081 (0.004)	0.264 (0.007)	0.68 (0.011)	0.26 (0.007)
6	915	503	54.97	1.42 (0.015)	0.075 (0.003)	0.259 (0.007)	0.68 (0.009)	0.26 (0.007)
7	1066	580	54.41	1.48 (0.013)	0.079 (0.003)	0.295 (0.006)	0.72 (0.009)	0.29 (0.006)
8	823	459	55.77	1.38 (0.014)	0.079 (0.004)	0.246 (0.007)	0.67 (0.011)	0.25 (0.007)
9	762	437	57.35	1.34 (0.014)	0.076 (0.004)	0.221 (0.007)	0.66 (0.013)	0.22 (0.007)
10	1110	538	48.47	1.47 (0.014)	0.088 (0.003)	0.289 (0.007)	0.69 (0.009)	0.29 (0.007)
11	1110	510	45.95	1.42 (0.014)	0.096 (0.004)	0.265 (0.007)	0.64 (0.011)	0.26 (0.007)
UN	1449	461	31.82	1.37 (0.015)	0.086 (0.005)	0.234 (0.008)	0.59 (0.016)	0.32 (0.008)
Overall mean	11940	5864	49.11	1.43 (0.004)	0.084 (0.001)	0.267 (0.002)	0.67 (0.003)	0.27 (0.002)

Note: %P = Polymorphism, NPS = number of polymorphic SNPs, NSU = number of SNPs used, N_e_ = number of effective alleles per locus, H_o_ = observed heterozygosity, uH_e_ = unbiased gene diversity, F_IS_ = inbreeding coefficient, PIC = polymorphic information content, UN = unknown, the values within the brackets are standard error.

**Table 2 plants-11-03480-t002:** Genetic diversity within and among the 90-cowpea accession classified by growth habit.

Pop.	Na	Ne	I	Ho	He	F_IS_	%P	P_A_
	Model-based population structure analysis							
SP1	5	1.189 (0.006)	0.189 (0.004)	0.035 (0.002)	0.162 (0.003)	0.684 (0.007)	30.68	3
SP2	18	1.368 (0.004)	0.352 (0.003)	0.116 (0.002)	0.233 (0.002)	0.398 (0.005)	80.64	229
SP3	40	1.397 (0.004)	0.381 (0.003)	0.083 (0.001)	0.247 (0.002)	0.617 (0.004)	90.96	508
SP4	27	1.258 (0.004)	0.257 (0.003)	0.068 (0.001)	0.165 (0.002)	0.437 (0.005)	68.66	42
Overall	90	1.303 (0.002)	0.295 (0.002)	0.075 (0.001)	0.202 (0.001)	0.514 (0.003)	67.74	-
	Distance-based population structure analysis							
C1	41	1.342 (0.004)	0.340 (0.003)	0.059 (0.001)	0.218 (0.002)	0.682 (0.004)	86.77	65
C2	24	1.404 (0.004)	0.387 (0.003)	0.108 (0.001)	0.254 (0.002)	0.512 (0.005)	88.93	234
C3	25	1.385 (0.004)	0.366 (0.003)	0.097 (0.001)	0.240 (0.002)	0.526 (0.005)	85.44	177
Overall	90	1.377 (0.003)	0.364 (0.002)	0.088 (0.001)	0.237 (0.001)	0.573 (0.003)	87.05	-

Note: Na = average number of observed alleles per locus per subpopulation, Ne = average number of effective alleles per locus per subpopulation, I = Shannon information index, Ho = observed heterozygosity per subpopulation, He = expected heterozygosity per subpopulation, FIS = inbreeding coefficient, %P = percentage of polymorphic loci, PA = private alleles, the values within the brackets are standard error.

**Table 3 plants-11-03480-t003:** Analysis of molecular variance (AMOVA) among 90-cowpea accessions classified based on SNP markers.

Source	DF	SS	MS	Est. Var	Per. Var	F-Statistics
**Model-based structure analysis**						
Among Population	3	37,575.3	12,525.1	270.2	22.14	F_ST_ = 0.221 (*p* < 0.001)
Among Individual	86	144,053.5	1675.0	725.2	59.44	F_IS_ = 0.763 (*p* < 0.001)
Within Individual	90	20,217.5	224.6	224.6	18.42	F_IT_ = 0.816 (*p* < 0.001)
Total	179	201,846.2	-	1220.0	100.00	-
**Distance based structure analysis**						
Among Population	2	20,380.1	10,190.1	143.8	12.16	F_ST_ = 0.122 (*p* < 0.001)
Among Individual	87	161,248.8	1853.4	814.4	68.85	F_IS_ = 0.784 (*p* < 0.001)
Within Individual	90	20,217.5	224.64	224.6	18.99	F_IT_ = 0.810 (*p* < 0.001)
Total	179	201,846.4	-	1182.8	100.0	-

Note: DF = degrees of freedom, SS = sum of squares, MS = mean sum of squares, Est. Var = estimated variance, Per. Var = percentage variation.

**Table 4 plants-11-03480-t004:** Pairwise estimates of gene flow (above diagonal, within the brackets), genetic differentiation (FST) (above diagonal off brackets), genetic distance (GD) (lower diagonal off brackets), and genetic identity (GI) (lower diagonal within the brackets).

	SP1	SP2	SP3	SP4
SP1	-	0.239 (0.796)	0.142 (1.511)	0.192 (1.052)
SP2	0.214 (0.807)	-	0.106 (2.108)	0.149 (1.428)
SP3	0.112 (0.894)	0.103 (0.902)	-	0.103 (2.172)
SP4	0.134 (0.874)	0.135 (0.874)	0.089 (0.915)	-

## Data Availability

Not available.

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
