# Peer review of "Analysis of Genetic Diversity and Population Structure of Cowpea (*Vigna unguiculata* (L.) Walp) Genotypes Using Single Nucleotide Polymorphism Markers"

_plants, 2022, doi:10.3390/plants11243480_

Round 1

Reviewer 1 Report

There are some comments within the PDF file revised which may help you to improve the article, including some word mistakes and english corrections.

Reviewer 2 Report

I forwarded my detailed questions and comments in the attached document. Authors should critically consider them, particularly on the methods and discussion part and some minor comments on the intro and result parts.

Reviewer 3 Report

Overall, this is an excellent paper. I would recommend acceptance with some edits and revisions. In line 232, how were the protocols of Nei and Li implemented?  Was it an R program or GenAlex?  Similarly, how was the bootstrap implemented from line 236? In line 219, you describe an in house pipeline that is "based on algorithms". This algorithm needs to be clarified or the wording needs to be edited to sound better. The supplemental data is highly deficient and must be improved. First, the supplemental data only contains 45 of the 90 genotypes. Also, the "origins" column is printed twice with different answers depending on the column. This data may be supplemental, but I still feel it is very important for showing the breadth of genotypes that were surveyed. In general, there are numerous places in the text where the English needs improvement. It is fairly minor but still highly noticeable to someone who is a native
English speaker, such as myself. I have specific lines in mind, but the numbering is missing from about half the copy that I received. The numbering starts with the section entitled, "Population structure and clustering".  Problem English is found at the following lines:  line 57 exited = existing, line 81 implement = implementing, line 106 population = populations, lines 156-158, lines 161-164, line 208 -80C needs a proper degree mark, lines 259-260. This is not an exhaustive list of English usage and grammar issues. It needs some work. 

Round 2

Reviewer 1 Report

Dear authors,

I still feal that Figure2 and 3 can be improved for a better view and understanding.

You answered in the first report that boostrap values makes the picture looks very untied. I do not really understand why? If you plot only those above 50 it won´t disturb, but improve the strenght or not of the main branches.

Why did you include K=2 and K=3 ? Explain it in the text.I think another reviewer suggested to include K=1 and K=2 in figure 2A, not 2B.

You answered also: "Individuals with a probability score of above 80% for a given cluster were considered as ‘pure’, whereas those with less than 80% were labelled as ‘admixture’". You may restrict the percentage of pure genotypes to 90% or even more. 80% it is low, 20% on introgression from other genome makes that genotype not to be pure, but admixed.

Please, check other suggestions within the PDF file.

Reviewer 2 Report

Comments addressed. 
